Corrected: Author correction

# Muscle pathology from stochastic low level DUX4 expression in an FSHD mouse model

Darko Bosnakovski[1,2,3], Sunny S.K. Chan[1,2], Olivia O. Recht[1,2], Lynn M. Hartweck[1,2], Collin J. Gustafson[1,2], Laura L. Athman[1,2], Dawn A. Lowe[4] & Michael Kyba [1,2]

Facioscapulohumeral muscular dystrophy is a slowly progressive but devastating myopathy caused by loss of repression of the transcription factor DUX4; however, DUX4 expression is very low, and protein has not been detected directly in patient biopsies. Efforts to model DUX4 myopathy in mice have foundered either in being too severe, or in lacking muscle phenotypes. Here we show that the endogenous facioscapulohumeral muscular dystrophy-specific DUX4 polyadenylation signal is surprisingly inefficient, and use this finding to develop an facioscapulohumeral muscular dystrophy mouse model with muscle-specific doxycycline-regulated DUX4 expression. Very low expression levels, resulting in infrequent DUX4 + myonuclei, evoke a slow progressive degenerative myopathy. The degenerative process involves inflammation and a remarkable expansion in the fibroadipogenic progenitor compartment, leading to fibrosis. These animals also show high frequency hearing deficits and impaired skeletal muscle regeneration after injury. This mouse model will facilitate in vivo testing of therapeutics, and suggests the involvement of fibroadipogenic progenitors in facioscapulohumeral muscular dystrophy.

[1] Lillehei Heart Institute, University of Minnesota, Minneapolis, MN 55455, USA. [2] Department of Pediatrics, University of Minnesota, Minneapolis, MN 55455, USA. [3] Faculty of Medical Sciences, University Goce Delcev - Stip, Stip 2000, Macedonia. [4] Division of Rehabilitation Science and Division of Physical Therapy, Department of Rehabilitation Medicine, University of Minnesota, Minneapolis, MN 55455, USA. Correspondence and requests for materials should be addressed to M.K. (email: kyba@umn.edu)

Facioscapulohumeral muscular dystrophy (FSHD) is one of the most prevalent muscle diseases, affecting an estimated half million individuals[1]. The disease is dominant, caused by mutations disrupting epigenetic silencing of the D4Z4 macrosatellite repeat at 4qter[2–6]: usually reductions in D4Z4 copy number[7], but also second site mutations in epigenetic regulators[6, 8]. Disrupted silencing on a specific allelic background, which provides a downstream poly A signal[9], leads to expression of DUX4, whose ORF is embedded in each D4Z4 repeat and encodes a double homeodomain transcription factor[9–11], and to activation of DUX4 target genes[12] through a mechanism involving p300/CBP[13].

Although DUX4 target genes are detectable in FSHD biopsy specimens[14], the DUX4 protein itself has not been detected by immunostaining, meaning that its expression is either very low; or stochastic, infrequent and possibly followed by cell death. In cultured FSHD myoblasts, DUX4 can be detected in at most 1/1000 cells[15, 16], but is more frequent upon differentiation into myotubes[17]. Forced high-level DUX4 expression causes death of myoblasts in vitro, while low-level expression prevents their differentiation[18].

The lack of a genetic mammalian model in which to study DUX4-mediated muscle pathology, and to test therapeutic interventions targeting DUX4, is currently a major roadblock in understanding of and developing treatments for FSHD. DUX4/D4Z4 transgenic mice have to date either lacked a muscle phenotype[19], or displayed such a strong multisystem phenotype that most die as embryos and rare live-born animals die before adult muscle develops[20]. The latter mouse, named iDUX4[2.7] carried a doxycycline-inducible DUX4 gene followed by the SV40 poly A signal integrated onto the X chromosome. Females showed severely biased X-inactivation of the DUX4-bearing X, presumably due to the competitive advantage that DUX4-X-inactivated cells have over those that inactivate the wild-type (WT) X during embryogenesis, thus females propagated the strain but were not themselves suitable for analysis. In the current study, we find that by changing the polyadenylation signal used in this X-linked transgene, basal expression in the off state is reduced and the multisystem phenotype is ameliorated. This new mouse model now enables growth to adulthood and the study of muscle-specific expression of DUX4 in both males and females. We find that very low levels of DUX4 expression in skeletal muscle fibers lead to an expansion in the fibroadiopgenic progenitor compartment, and to a progressive fibrotic degeneration of skeletal muscles.

## Results

### The endogenous DUX4 polyadenylation signal is inefficient.
FSHD only occurs when the derepressed D4Z4 array is present upstream of a permissive (4qA) allele, the defining feature of which is a polyadenylation signal that is thought to provide stability to the message and thereby allow translation of the DUX4 protein[9]. In the iDUX4[2.7] mouse, this sequence was present together with an additional SV40 poly A (Fig. 1a). Because read through transcription of the DUX4 poly A in the iDUX4[2.7] model would be stabilized by the SV40 poly A, measuring mRNA upstream and downstream of the 4qA poly A in this context is a sensitive way of quantifying its activity. We discovered significant read through of the endogenous DUX4 pA (Fig. 1b), and hypothesized that the SV40 pA may provide disease-inappropriate stability to the DUX4 message in the iDUX4[2.7] mouse. Therefore, we generated a doxycycline-inducible DUX4 model lacking the SV40pA. A genomic fragment from the terminal D4Z4 repeat of an FSHD 4qA161 allele including the DUX4 ORF, 3′ UTR and DUX4 pA was introduced at the same location

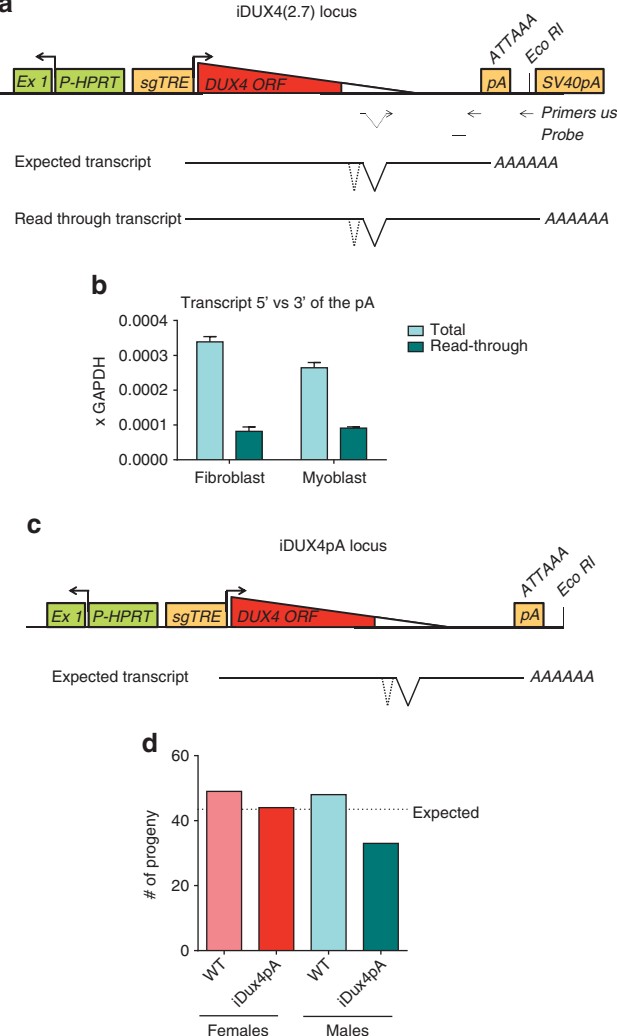

**Fig. 1** DUX4 poly A signal read through and the effect of removing the SV40 pA signal. **a** Schematic of transgene integration in iDUX4[2.7] mice and primers/probe used for detection of transcript proximal to and distal of the DUX4 pA signal. Note the SV40 poly A signal downstream of the DUX4 poly A signal. **b** RTqPCR with reverse primers on either side of the DUX4 pA signal in PDGFRα⁺ integrin α7⁻ (fibroadiopgenic) and PDGFRα⁻ integrin α7⁺ (myogenic) sorted cultured primary muscle cells. $n = 3$ technical replicates. Note the large amount of transcript that is detected by the primer downstream of the DUX4 pA, indicating that the DUX4 poly A is not efficient and allows a high level of read through. In the iDUX[2.7] mouse, these read through transcripts are stabilized by the SV40 poly A, potentiating higher basal levels of DUX4 in the off state. **c** The transgene is targeted 5′ of HPRT, is regulated by a second generation tetracycline-response element, includes DUX4 and 3′ UTR including the DUX4 poly A signal. **d** Mice were made in which the SV40 pA was removed, leaving only the endogenous, less efficient, DUX4 pA. Viability is now near-normal for male carriers. Summary of 174 genotypes of 3-week old mice from 21 litters of iDUX4pA females backcrossed to males with a WT chrX. Results are not significantly different from expected for the hypothesis of no selection against iDUX4pA ($\chi^2$, two tailed $p = 0.2956$), but are strongly diverged from expected ($p < 0.0001$) for the hypothesis that iDUX4pA is selected against at the same level as observed against iDUX4[2.7][20]. Thus, we conclude that the presence of the DUX4 poly A makes the transgene less toxic, with viability to 3 weeks not different from WT in the absence of dox

upstream of *HPRT* under the control of the doxycycline-inducible promoter. We refer to this strain as iDUX4pA (Fig. 1c). Unlike the previous iDUX4[2.7] line in which live-born carrier males are rare and die as weanlings, iDUX4pA males survive to 3 weeks at near-normal ratios (Fig. 1d).

**Muscle phenotypes in the absence of doxycycline**. Carrier males grow well, but have a slightly reduced body weight and live up to 4 months, while females appear normal (Fig. 2a). Like the iDUX4

[2.7] strain, the iDUX4pA mice have skin hyperkeratosis and alopecia. Males have much reduced body fat (Supplementary Fig. 1a), although they eat normally. Male muscles appeared atrophic, and proportional to body weight, were significantly smaller (Fig. 2b), more obvious if normalized to lean body weight since these animals have little fat (Supplementary Fig. 1b). Although capable of normal ambulation, males were less active overall (Supplementary Fig. 1c), and showed significant defects on functional strength and locomotor measurements (Fig. 2c). The contractile strength of the isolated extensor digitorum longus

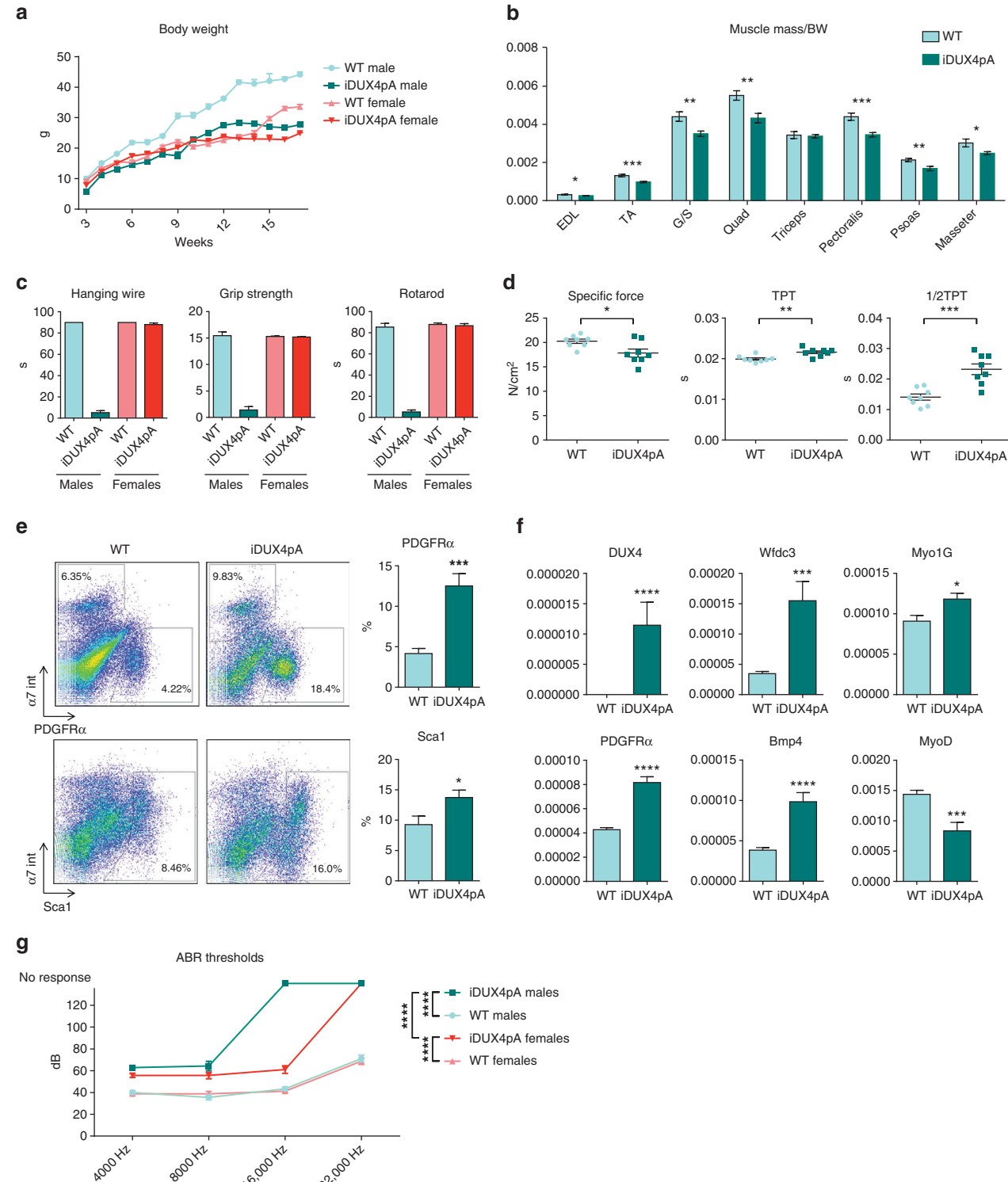

muscle (EDL) was significantly diminished, and it showed lower specific isometric force significantly decreased rate of contraction and extended relaxation time (Fig. 2d, Supplementary Fig. 2a). The muscle did not show obvious histological signs of dystrophy, however analysis of cellular composition by flow cytometry revealed a large increase in Lin[neg] (CD45[neg] and CD31[neg]) integrin α7[neg] PDGFRα[+] presumptive fibroadipogenic progenitors (FAPs)[21, 22] (Fig. 2e, Supplementary Fig. 3a, b). We did not detect a significant change in the frequency of Lin[neg] integrin α7[+] VCAM[+] myogenic progenitors or CD45[neg] CD146[+] CD31[+] pericytes (Supplementary Fig. 3c, d). However, small but significant alterations of CD45[neg] Sca1[+] CD31[+] endothelial progenitors[23] and CD206[+] macrophages were detected (Supplementary Fig. 3e, f). DUX4 mRNA could be detected at extremely low levels by RTqPCR and its targets Myo1g and Wfd3c were elevated (Fig. 2f, Supplementary Fig. 4). Additional transcriptional analyses in muscle revealed induction of some genes involved in fibrosis and reduced expression of MyoD (Fig. 2f). Extremely low DUX4 basal expression together with elevated levels of DUX4 targets were also detected in some non-muscle tissues in males (Supplementary Fig. 4). iDUX4pA females presented milder skin phenotypes, and did not display muscle atrophy or decreased functional strength or muscle contractile force (Fig. 2c, Supplementary Fig. 2b).

**Evaluation of high frequency hearing loss**. High frequency hearing loss is a non-muscle phenotype associated with FSHD[24, 25]. We therefore evaluated the auditory brainstem response (ABR) of iDUX4pA mice to sounds of increasing amplitude across a range of frequencies. This revealed that both male and female iDUX4pA mice are hearing-impaired, particularly at the higher frequency ranges, with males being more severely affected and deaf to sounds of 16 kHz and above (Fig. 2g).

**Muscle-specific DUX4 induction leads to dystrophic changes**. To evaluate DUX4 effects in muscle, we first investigated the effect of DUX4 expression on primary cultured cells from muscle using the ubiquituously expressed Rosa26-rtTA2sM2 driver[26]. As previously demonstrated, DUX4 displayed cytotoxicity to both myogenic (VCAM1[+] integrin α7[+]) and fibroadipogenic (integrin α7[neg] PDGFRα[+]) progenitors when induced to high levels of expression (Supplementary Fig. 5), and impaired myogenic differentiation at low levels of expression (Supplementary Fig. 6). With the ubiquitous rtTA, animals died within 24 h of dox treatment, therefore to evaluate the effects of DUX4 induction on muscle in vivo, we replaced the Rosa26-rtTA driver with the skeletal muscle-specific HSA-rtTA driver[27]. Exposure of 6-week-old male mice to doxycycline led to significant reductions in muscle size in a dose- and time-dependent manner (Fig. 3a, b). Muscles showed differential sensitivity, with gastrocnemius/soleus and quadriceps being highly sensitive and TA muscle less

so. Remarkably, similar sensitivities were observed in females, whose muscles are normal in size prior to induction (Supplementary Fig. 7a). Doxycycline had no effect on muscle size of mice lacking HSA-rtTA. We measured force-generation capacity of the isolated EDL and found absolute force reduced by ~50% after 28 days of low (5 mg/kg) dox, with significantly decreased specific tetanic, isometric and concentric force (Fig. 3b, Supplementary Fig. 8).

We investigated DUX4 expression and found that most fiber nuclei were negative, but sporadic nuclei were DUX4[+] with varying staining intensities (Fig. 3c). Expression was not observed in interstitial cells, and positive nuclei were generally surrounded by MHC + cytoplasm (Supplementary Fig. 7b). DUX4 mRNA and upregulation of DUX4 target genes were robustly detected in muscle but not in liver (Fig. 3d).

Coincident with these physiological changes, we observed dystrophic changes to muscle: excessive heterogeneity of fiber size including many small fibers, damaged and necrotic fibers, mononuclear cell infiltrate, a large excess of extracellular matrix deposition between fibers and some central nucleation, in both males and females (Fig. 3e, Supplementary Fig. 7c). This situation is strongly reminiscent of FSHD patient biopsies, where histological signs are obvious, the DUX4 target gene expression profile is detectable, but the DUX4 protein has proven very difficult to identify by immunohistochemistry.

We quantified the fibrotic changes in the gastrocnemius, and found that extremely high induction (100 mg/kg dox) led to ~15% of muscle cross-sectional area turning over to fibrotic tissue within 14 days (Fig. 3e). We also observed an increase in mRNAs associated with a fibrotic program, including TGFβ1 and Col1a1 (Fig. 3f). We did not detect significant increases of PDGFRα[+] and Sca1[+] cells with DUX4 induction (Supplementary Fig. 8), most likely, because those progenitors were already very high in iDUX4pA males compared to WT (Fig. 2f, Supplementary Fig. 3b). In females on the other hand, where FAP frequencies were much lower prior to induction, a significant induction of FAPs was detected in response to DUX4 in a dose- and time-dependent manner (Supplementary Fig. 10).

Inflammatory cells can stimulate the pro-fibrotic activity of FAPs[28], therefore we evaluated muscle by flow cytometry for markers of inflammatory myeloid cells. This revealed a large increase in frequency of CD45 cells expressing the markers Ly6G, CD68, and CD206 after dox induction, consistent with this notion (Fig. 3g, Supplementary Fig. 11).

**DUX4 induction severely impairs regeneration after injury**. Because DUX4 expression inhibits myogenesis, it has been proposed that impaired regeneration after normal wear and tear or injury could contribute to muscle loss in FSHD[18]. To test the effect of DUX4 on skeletal muscle regeneration after injury, we damaged TA muscles of HSA-rtTA; iDUX4pA male and female

**Fig. 2** Phenotypes of iDUX4pA mice in the basal "off" state. **a** Weekly average of the weight of iDUX4 and wild type (WT) male and female siblings (n = 3–25). **b** Normalized mass to the body weight (BW) of various muscles in iDUX4pA males. Extensor digitorum longus (EDL), tibialis anterior (TA), gastrocnemius and soleus (G/S), quadriceps (Quad), triceps brachii (Triceps), pectoralis, psoas major (Psoas), and masseter from 10 to 15 week old mice and age-matched siblings are analyzed (n = 10). **c** Representative example of hanging wire, grip strength and rotarod analyses of 6-week-old iDUX4pA male and female mice (n = 4). **d** Specific force, time to reach peak twitch force (TPT) and time to reach 1/2 relaxation during twitch (1/2 TPT) of EDL in 6–10-week-old iDUX4pA males and age-matched siblings (n = 8). **e** Representative FACS analyses for Lin[neg] (CD45[neg]; CD31[neg]) and PDGFRα[+] or Sca1[+] cells in muscle from iDUX4pA males and sibling controls. Percent of PDGFRα[+] or Sca1[+] cells in pooled muscle digests comprising TA, gastrocnemius, soleus, quadriceps, pectoralis, and triceps is shown at right (n = 7). **f** RTqPCR analyses for DUX4, DUX4 targets, myogenic genes, and genes involved in fibrosis in pectoralis from iDUX4pA male mice. Results are presented as fold difference to GAPDH (n = 4). **g** Auditory brainstem response (ABR) in iDUX4pA males, females and WT sibling controls (n = 5, 6–8 week old mice). Note that iDUX4pA males do not respond to sounds of 16 kHz and above and iDUX4pA females to sounds of 32 kHz. All data are presented as mean±SEM; *p < 0.05, **p < 0.01, ***p < 0.001, ****p < 0.0001 by T-test or **g** two-way ANOVA followed by Tukey's post hoc test

animals by cardiotoxin injection. In the presence of dox to induce DUX4, 1 month post-injured muscles showed only small fibers, together with severe fibrosis in the zone of injury (Fig. 4a, Supplementary Figs. 12, 13). Immunostaining revealed stochastic but more prominent DUX4 expression in cells in the injured area, compared to differentiated fibers in the uninjured area. In the mouse, cardiotoxin-injured TA muscles undergo mild hypertrophy to recover to slightly greater than pre-injury mass within 3 weeks. We found that this did not occur in the presence of DUX4, rather we saw instead a large and dose-dependent loss

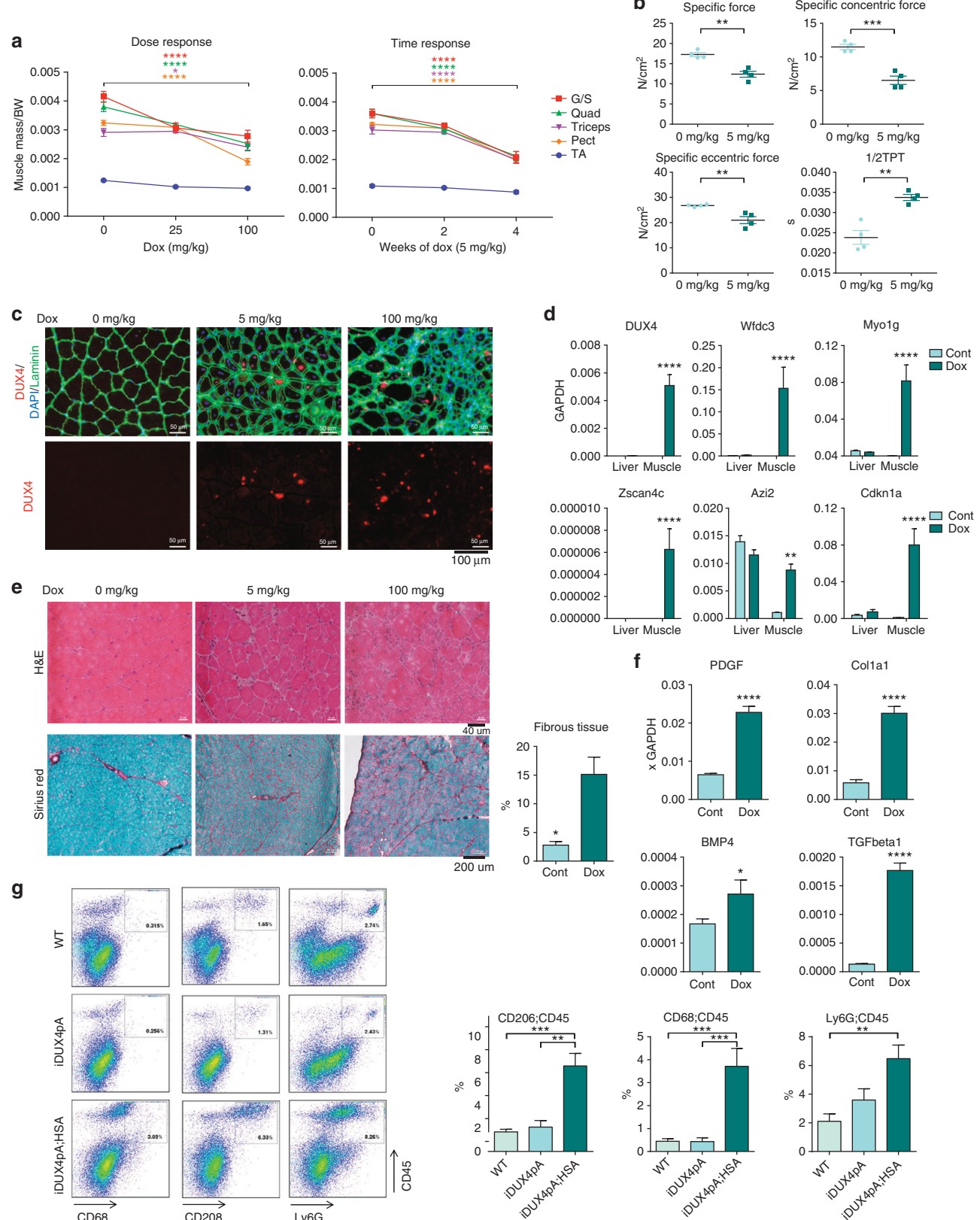

of muscle mass (Fig. 4b–d). Thus, not only does DUX4 expression in established fibers lead to dystrophy, but DUX4 expression in differentiated myogenic cells in the context of regeneration leads to profoundly severe dystrophic changes.

## Discussion

The DUX4 poly A signal is ATTAAA, which is more subject to alternative usage than the more common AATAAA[29], which may explain why it allows significant read-through. By making DUX4 expression dependent on its endogenous relatively inefficient pA, we have reduced the basal off-state activity and eliminated the dominant lethality of the inducible *DUX4* transgene. With a ubiquituous rtTA driver, mice die within 24 h of dox treatment, however with the skeletal muscle-specific HSA-rtTA, mice survive beyond 5 weeks without obvious phenotypes outside of the muscle, demonstrating the tightness of the system. The skeletal muscle pathologies acquired upon doxycycline induction demonstrate a remarkable and important similarity to the most unusual aspect of FSHD: in human muscle biopsies, the DUX4 protein is virtually invisible by immunohistochemistry, although its RNA and target gene signature can be detected. Likewise, the DUX4 target gene signature is detectable at the RNA level in the uninduced iDUX4pA mouse, but the DUX4 protein is not visible by immunohistochemistry. This very low expression may be due to the low efficiency of the inducible system in skeletal muscle fibers, and/or to the unexpected inefficiency of the endogenous DUX4 4qA161 poly A signal. Nevertheless, muscles are wasted, proportionally smaller and weaker. After induction, DUX4 is only expressed in a small minority of myonuclei, yet this increase in expression leads to devastating dystrophic changes over a period of several weeks.

The mice also exhibit another notable FSHD-associated phenotype—high frequency hearing loss. This phenotype occurs in the absence of induction, i.e., with low, protein-undetectable levels of DUX4 from the basal "off" state of the dox-inducible promoter.

The iDUX4pA model is both conditional and titratable, thus both the timing of onset of muscle loss and its severity can be controlled. At low induction levels, chronic atrophy leading to muscle weakness ensues, while at high induction levels, more severe muscle damage occurs leading to loss of ambulation. This titratability is particularly relevant to FSHD as the disease presents with wide variability in both severity and age of onset. Although the iDUX4pA transgene is X-linked, induced females showed robust loss of muscle function with no greater variance than males. We attribute this to the granularity of X-inactivation combined with the large number of myonuclei per fiber. This leads to the likelihood that most fibers contain a subpopulation of nuclei with DUX4-bearing active X chromosomes. The robust ability to induce phenotypes at FSHD-relevant levels of DUX4

expression in both males and females is well-suited to testing therapeutics targeting the DUX4 protein or mRNA.

Notwithstanding the striking similarities to FSHD described above, the iDUX4pA mouse does not show certain hallmarks of the disease, such as the spatial pattern of affected muscles. These are partly by design (phenotypes are limited by the spatial activity of the conditional promoter used), but may also relate to the fact that a large set of DUX4 target genes are regulated in humans but not mice[30]. However, given the striking phenotypes observed, those targets that are in common between the two species must explain the major component of pathology, with differences possibly explaining the subtleties.

The pathological sequence by which DUX4 leads to muscle wasting in FSHD is currently enigmatic. Previous FSHD transgenic mouse models have not been useful as they have shown no muscle phenotypes[19, 20] or have overexpressed the wrong gene[31]. However, relevant disease mechanisms can now be probed. We found that although muscle of iDUX4pA males showed no signs of frank dystrophy prior to induction, it harbored an excess of PDGFRα⁺ FAPs, suggesting that the lowest levels of DUX4 expression set up a profibrotic state. After doxycycline treatment to stochastically induce IHC-visible levels of DUX4 in fibers, muscles experience an influx of inflammatory cells and the initiation of a fibrotic program. Since DUX4 induction was confined to myofibers, this effect is most likely non-cell autonomous. Muscle tissue is lost, and myogenesis increases, but myogenesis clearly cannot compensate productively, with the end result being severe loss of force-generation capacity. We directly tested the effectiveness of myogenesis by cardiotoxin injury and found that it was profoundly impaired in both males and females. The ability to combine with different rtTA drivers will allow the iDUX4pA model to be used to investigate potential pathological roles for DUX4 expression in myogenic progenitors and other cells of muscle. The involvement of FAPs in response to DUX4 expression in muscle clearly warrants considering non-cell autonomous mechanisms in FSHD[32] and investigations into FAPs in FSHD biopsy material.

The absence of a suitable animal model has to date severely limited both investigations into disease mechanisms and investment in pharmacological therapies for FSHD. The iDUX4pA mouse, with disease-relevant low-level and stochastic DUX4 expression, and conditional titratable muscle phenotypes will facilitate these efforts.

## Methods

**Targeting construct cloning and generation of iDUX4pA mice**. The SV40 pA signal was deleted from p2Lox-DUX4[20] and the resulting plasmid introduced by electroporation into ZX1 inducible cassette exchange mouse ES cells[33]. Chimeric animals were derived at the University of Minnesota Mouse Genetics Laboratory Core Facility by blastocyst injection. Mice were maintained under a protocol approved by the University of Minnesota IACUC. These mice have been assigned Stock No. 030749 (Jackson Labs).

---

**Fig. 3** DUX4 induces skeletal muscle atrophy. **a** Mass of various muscles normalized to the body weight (*BW*) of iDUX4pA male mice treated with doxycycline over doses and time ranges ($n = 3$). **b** Specific isometric, concentric and eccentric forces and 1/2 relaxation time (1/2 TPT) of EDL in 10-week-old iDUX4pA male mice induced with 5 mg/kg doxycycline for 28 days ($n = 4$). **c** Immunostaining for DUX4 and laminin on quadriceps of iDUX4pA;HSA-rtTA males induced with 5 and 100 mg/kg doxycycline for 14 days. DAPI was used for nuclear staining. **d** RTqPCR analyses for *DUX4* and DUX4 target genes *Myo1g, Wfdc3, Azi2, Cdkn1a,* and *Zscan4c* in pectoralis muscle and liver of iDUX4pA;HSA-rtTA male mice induced with 25 mg/kg doxycycline for 14 days. Results are presented as fold difference to *GAPDH* ($n = 3$). **e** Hematoxylin and Eosin (*H&E*) staining and Sirius red/fast green staining of quadriceps from iDUX4pA;HSA-rtTA male mice induced with 5 and 100 mg/kg doxycycline for 14 days (*left*). Quantification of fibrous tissue in control and induced (100 mg/kg) quadriceps ($n = 3$, *right*). **f** RTqPCR for indicated fibrogenic markers in iDUX4pA;HSA-rtTA males mice induced with 5 mg/kg doxycycline for 14 days. Results presented as fold difference to *GAPDH* ($n = 3$). **g** Representative FACS analyses for hematopoietic inflammatory cells expressing CD45 and CD68, CD206 or Ly6G in muscle of iDUX4pA;HAS-rtTA males or control siblings (WT or lacking rtTA) induced with 25 mg/kg doxycycline for 14 days (*left*), and summary data (*right*, $n = 4$). Data are presented as mean±SEM; *$p < 0.05$, **$p < 0.01$, ***$p < 0.001$, ****$p < 0.0001$ by *T*-test except **a**, **b** two-way ANOVA or **g** one-way ANOVA, with Tukey's post hoc test

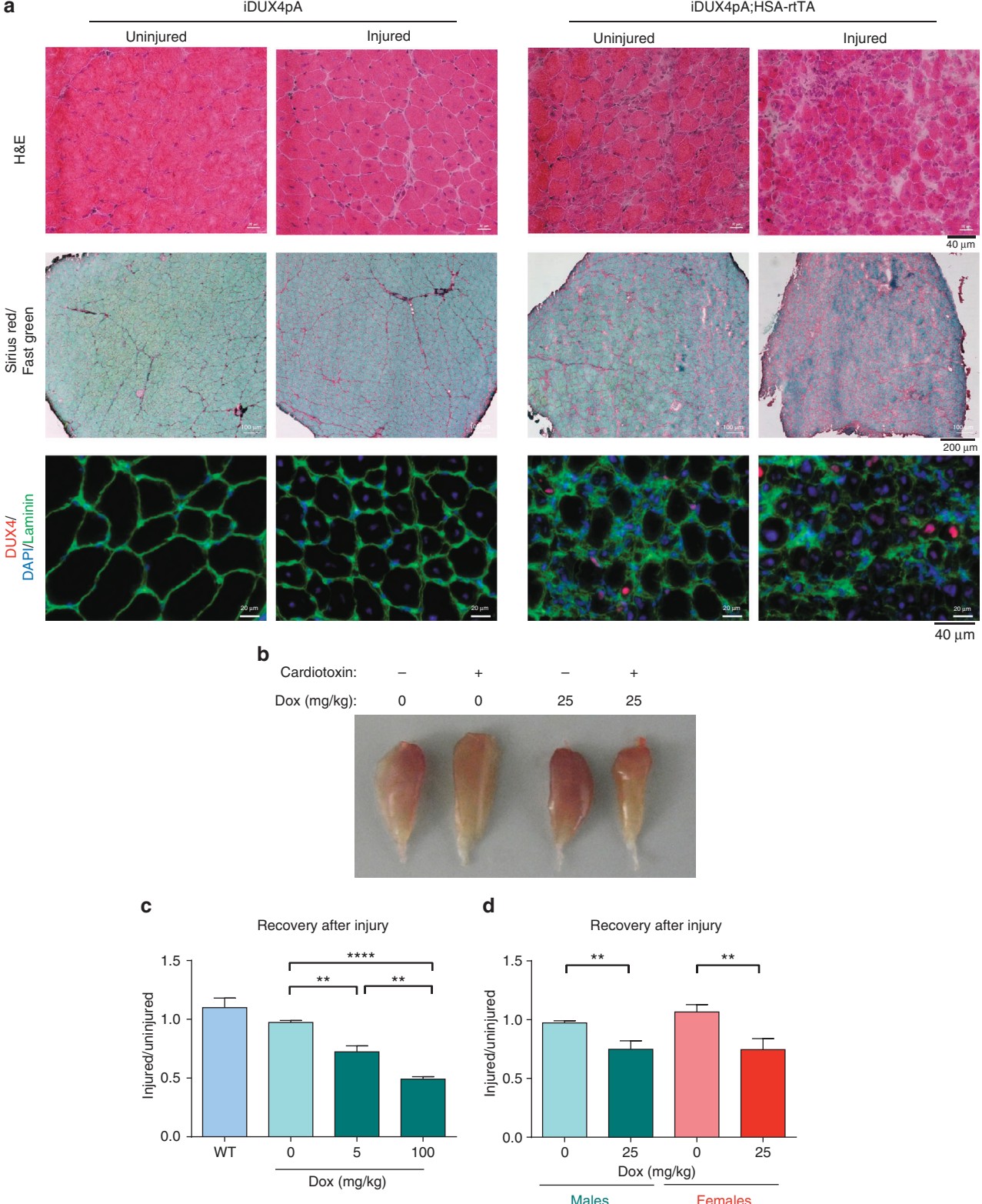

**Fig. 4** DUX4 impairs recovery after injury. **a** Hematoxylin and eosin (*H&E*), Sirius red/fast green and immunostaining for DUX4, Laminin and DAPI nuclear staining of uninjured and injured TA muscle sections from iDUX4pA;HSA-rtTA males and controls lacking rtTA 12 days post injury. Mice were induced with 25 mg/kg doxycycline for 12 days. **b** Gross morphology of injured/uninjured TA muscles with/without DUX4 induction 12 days post injury. **c** Ratio of weight of post injury vs. uninjured contralateral TA from iDUX4pA;HSA-rtTA males, and siblings lacking rtTA or fully WT 12 days post injury ($n = 3$). **d** Ratio of post injury vs. uninjured contralateral TA from iDUX4pA;HSA-rtTA male and female mice 12 days post injury. Mice were induced with 25 mg/kg doxycycline ($n = 3$). Data are presented as mean±SEM; **$p < 0.01$, ****$p < 0.0001$ by **c** one-way ANOVA with Tukey's post hoc test or **d** *T*-test

**RNA isolation and RTqPCR**. RNA was isolated from snap frozen muscles or cultured cells using Trizol (Invitrogen) according to the manufacturer's directions. In total, 3 μg RNA was treated with DNase (Promega), and cDNA was synthetized using SuperScript IV Reverse Transcriptase (Invitrogen). qPCR was performed using Premix Ex Taq (Probe qPCR) Master Mix (Takara) and commercially available TaqMan probes (Applied Biosystems, see Supplementary Table 1), except for *DUX4* which was detected using FAM-labeled probe (TCTCTGTGCCCTTG TTCTTCCGTGAA) and custom primers (PLH298-PAS-F: CCCAGGTACCAGCA GACC and PLH299-PAS-R: TCCAGGAGATGTAACTCTAATCCA, from[9]) detecting 3′ UTR sequences upstream of the pA signal or an alternative reverse primer (PLH300-DAS-R: TGATCACACAAAAGATGCAAATC) for detecting read through transcription. Expression analyses were performed on at least three biological and technical replicates.

**Isolated skeletal muscle contractile measurements**. Intact EDL muscles were dissected from anesthetized mice and placed in a tissue bath filed with Krebs-Ringer bicarbonate buffer. The distal tendon was statically attached to the bottom of the bath and the proximal tendon to a lever. Contractions were induced using a Grass S48 stimulator delivered through a SIU5D stimulus isolation unit (Grass Telefactor), and forces measured using a dual-mode muscle lever system (300B-LR; Aurora Scientific Inc.) and TestPoint software (Capital Equipment), as described in ref. [34].

**Body composition and measurement of mouse activity**. Cage activities were analyzed over a period of 72 h in open field activity chambers (Med Associates Inc., St. Albans, VT) and body composition was evaluated with Echo-MRI (Echo Medical Systems) at the IBP Phenotyping Core Facility, University of Minnesota.

**Hanging wire test**. The hanging wire test was adapted from Kondziela's inverted screen test. Mice were placed in the center of a framed wire mesh and the frame was inverted. The time that the mice clung to the wire was recorded. Scores for each mouse were averages of three tests.

**Grip strength test**. The weights test was done using seven different weights, ranging from 7 g to 98 g. Mice were suspended by the tail and allowed to grip with forepaws a tangled ball of fine gauge wire, attached to the lowest weight. Mice were moved to a greater weight if they held the tested weight three times for 3 s, with a 10 s break between each trial. The final score was the greatest weight held three times for 3 s.

**Rotarod test**. The rotarod test was run using a Rota Rod Rotamex 5 (Columbus Instruments). The rotarod was programmed to start at 10 rpm, and increased 5 rpm every 30 s. Average length of time spent on the rotarod from three replicates was recorded.

**Isolation and FACS analyses of mononuclear cells from muscle**. Pooled or individual muscles (TA, gastrocnemius, soleus, quadriceps, pectoralis, triceps) were minced and digested with collagenase type II and dispase. Levels of muscle infiltration with hematopoietic cells and FAPs were evaluated using FACS and following antibodies CD45, CD31, CD206, CD68, Ly6G, PDGFRα, Sca1, VCAM1, CD11b, CD144, CD146, and integrin α7 (Supplementary Table 2). Staining was done in PBS supplemented with 1% FBS on ice for 60 min. Samples were analyzed on a FACSAria and data analyzed using FlowJo (BD Biosciences). FACS gating strategy is shown in Supplementary Fig. 3a. All experiments were done on at least three biological replicates.

**Primary cultures of myogenic progenitors and FAPs**. Total muscle digests were plated in Ham's F-10 medium (Hyclone) supplemented with 20% fetal bovine serum (HyClone), 50 ng/μl human basic fibroblast growth factor (Peprotech), penicillin, streptomycin, and Glutamax (Gibco), and cultured at 37°C in 5% oxygen. After 5–7 days, myogenic progenitors were sorted out as CD45[neg] CD31[neg] integrin α7[+] VCAM1[+] and FAPs as CD45[neg] CD31[neg] integrin α7[neg] PDGFRα[+] cells using a FACSAria (BD Biosciences).

**Auditory evoked brainstem response analyses**. Hearing loss was done on ketamine/xylazine anesthetized mice on a BioSigRP TDT System 3 (Tucker-Davis Technology; Alachua, FL). Stimuli were presented as a Pure Tone and responses to 500 sweeps were averaged at each intensity level (frequencies from 4 kHz to 32 kHz and 10-dB SPL steps). The auditory threshold was defined as the lowest intensity level at which a clear ABR waveform was visible.

**Histology and immunofluorescence**. Muscles were embedded in OCT. All histological staining was done on 10 μm tissue sections. Fibrosis was visualized by Sirius red/fast green staining and the level of fibrosis determined using ImageJ Software[35]. Immunostaining was performed on 10% formalin fixed tissue sections or on cells cultured in 48-well plates. Samples were permeabilized with Triton-X, blocked with 3% BSA, incubated with primary antibody overnight, and

fluorochrome-conjugated secondary antibodies (Invitrogen) for 45 min. Nuclei were stained with DAPI (Sigma). Antibodies used for immunofluorescence are shown in Supplementary Table 2.

**Statistics**. To minimize selection bias, randomization was done by selecting pairs of animals of identical age and similar weight, and distributing between two groups blinded to group treatment. Experimenters were also blinded to group allocation during processing and analysis of samples. All replicates are biological, except where indicated. Sample sizes were selected based on effect size and availability of experimental/sibling control animals. Group data was tested for normality (Kolmogorov–Smirnov test) and differences between groups were evaluated by T-test (if not indicated), $\chi^2$ test, or two-way analysis of variance (ANOVA) followed by Tukey's or Sidak's post hoc tests as indicated, using Graphpad Prism software. Differences were considered significant at $p$-values of 0.05 or lower.

**Data availability**. The data that support the findings of this study are available from the corresponding author upon reasonable request.

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

## Acknowledgements

This work was supported by the National Institute for Arthritis and Musculoskeletal and Skin Diseases and National Institute on Aging grants RO1 AR055685, R01 AG031743 and by the Bob and Gene Smith Foundation. D.B. was supported in part by the Children's Cancer Research Fund. We thank Ning Xie for animal colony support. We thank Rita Perlingeiro and her laboratory for technical support.

## Author contributions

Conceptualization, M.K.; Methodology, M.K., D.B., and D.A.L.; Investigation, D.B., S.S.K.C., L.M.H., O.O.R., L.L.A., C.J.G., D.A.L.; Formal Analysis, D.B. and L.M.H.; Writing – Original Draft, M.K. and D.B.; Writing – Review & Editing, M.K., D.B., and D.A.L.; Visualization, M.K. and D.B.; Funding Acquisition, M.K. and D.A.L.; Supervision, M.K.

## Additional information

**Competing interests:** The authors declare no competing financial interests.

