## [Peer Review file · Nature Communications]

Reviewers' comments:

Reviewer #1 (Remarks to the Author):

Previous attempts at producing a mammalian animal model for FSHD have resulted in animals without a clear skeletal muscle phenotype or animals with a severe, multisystem lethal phenotype. Bosnakovski et al. present data on a transgenic mouse model of a doxycycline-inducible human DUX4 gene (iDUX4pA) integrated into the X chromosome. These transgenic mice survive into adulthood, have a slowly progressive myopathy aggravated by doxycycline induction and have high frequency hearing loss, an extra-muscular feature seen in patients with severe forms of FSHD. The authors claim that this inducible animal model closely mimics human FSHD, and will be important in exploring the mechanisms of DUX4 induced myopathy and is of relevance in FSHD drug development. The data provided in support of the conclusions are extensive and clear with enough detail to be reproducible.

The iDUX4pA animal model is the first potentially useful FSHD animal model and will be of great interest to the FSHD research community at a time when multiple therapeutic approaches are being actively explored. Although this study does not provide novel insights into disease mechanism, it does confirm experimentally, what earlier investigations have suspected, that DUX4 induction impedes normal muscle regeneration. Below are my specific comments:

- Although high frequency hearing loss is a feature of FSHD, it occurs in a small percentage of the patients with FSHD who have severe, early onset disease. The claim by the authors that high frequency hearing loss is highly prevalent and highly penetrant is inaccurate.
- FSHD muscle histopathology, except for the occasional occurrence of prominent inflammation, is not distinctive. The authors show that with doxycycline induction of a more active and destructive myopathic process, resulting in increased DUX4 mRNA as well as the ability to demonstrate DUX4 in rare myonuclei by IHC. These changes corresponded to a marked increase in Myo1g and Wfdc3, listed as DUX4 targets. However, I could not find references to either Myo1g or Wfdc3 as established DUX4 targets. It would have been very helpful if the authors demonstrated the presence of some of the other DUX4 targets previously established in human FSHD muscle (Yao et al., 2014)

Reviewer #2 (Remarks to the Author):

Bosnakovski et al present a transgenic mouse (iDUX4pA) with doxycycline-inducible DUX4 expression in skeletal muscles that closely mimics the pathological features of FSHD. This is a major achievement that clearly deserves publication in Nature Communications. Many attempts from several groups had failed up to now at generating an animal model of FSHD that could recapitulate the main characteristics of the human disease. The new iDUX4pA mice present tissue-specific DUX4 expression in skeletal muscles. This expression is tunable with doxycycline and can provide the wide level range observed in FSHD myoblast cultures where it is correlated with different DUX4 gene methylation levels.

There is a major interest in such a model because (i) it will allow for a better understanding of the FSHD pathological mechanism i.e. precisely deciphering the path between the molecular defect and the overt muscle (histo)pathology; (ii) it will provide an invaluable tool for screening or validating putative therapeutic agents (targeting DUX4 expression or pathways downstream of it) that are increasingly being sought for by researchers and pharmaceutical companies.

In the previous iDUX4[2.7] mouse model leaky ubiquitous DUX4 expression was highly toxic to male mice. Here, male iDUX4pA mice also show leaky DUX4 expression in the absence of inducer but its very low level strongly reduces the toxicity so they survive for at least 5 weeks.

The mouse characterization has been done thoroughly (weight, muscle function, histology, cell

sorting, RNA analyses for DUX4, its target genes, myogenic and fibrogenic genes; muscle regeneration).

Males in the absence of inducer have low leaky DUX4 expression, decreased muscle function with no muscle atrophy but associated with decreased myogenesis markers and increased fibroadipogenic progenitors. This point is of major interest since increased extracellular matrix deposition and fat infiltration are early events in FSHD muscles .

Both males and females have a skin phenotype which is not observed in patients but present high frequency hearing impairment that is typical of patients.

Dose- and time-dependent myopathy (atrophy) develops following DUX4 induction in all mice, with inflammatory infiltrates and fibrosis. This indicates that the selection against cells with the active X chromosome carrying the DUX4 transgene that was observed in the previous iDUX4[2.7] model does not occur here.

Muscle regeneration following cardiotoxin-mediated injury is impaired by DUX4 in a dose-dependent manner.

Could the Authors comment the following points?

- In non-induced mice where (in which tissues) does the leaky DUX4 expression occur? If indeed it is not possible to immunodetect DUX4 in tissue sections maybe detection of its footprint gene products could be done? (this experiment is not required for the present publication)
- What promoter could be used for this leaky expression? Is it linked to Sp1 binding sites where transcription could occur without a TATAA box?
- In doxycyclin-induced mice the HSA promoter leads to DUX4 expression after differentiation has been initiated thus not in proliferating precursor cells, so a pathogenic function of DUX4 at this early stage could not be detected in this mouse model. What is the nature of the cells with increased DUX4 expression in the regenerating muscles?
- Is there any evidence of DUX4 leaking from skeletal muscle cells to other cell types in this mouse model? What is the nature of the non-cell autonomous mechanism suggested in regeneration (discussion)? Could it be linked to the Cxcr4-Sdf1 axis as observed by Dmitriev et al 2016 Oncotargets?
- Previous studies have demonstrated differences in the gene deregulation cascade caused by DUX4 expression when studied in mouse or human cells. This is the case for genes with promoters derived from repeated elements that harbor DUX4 binding sites and are only present in human. In what respect would that explain pathological differences observed in the iDUX4pA mice and patients?

Minor comments:

1. Besides the epigenetic conditions the authors should mention in the introduction the genetic background required to develop FSHD i.e. a DUX4 gene with a polyadenylation signal (exon3 PAS) provided to the distal D4Z4 unit by the flanking pLAM sequence. This PAS is unusual in that its sequence is ATTA AAA (Dixit et al 2007) but not the standard AATAAAA as wrongly stated both in Figure 2a and Supplementary Fig 1. This should be stated in the beginning of the Result section and corrected in the Figures.
2. In the same figures: the first intron is in fact alternatively spliced (Dixit et al 2007; see review Anseau et al Genes 2017).
3. The authors have been stumbling for several years on the extreme DUX4 toxicity (even with no doxycycline induction) in their previous iDUX4 [2.7] mouse model, but not the readers, so could they please clarify the background! Lemmers et al 2010 demonstrated the exon3 PAS was required to get stable polyadenylated DUX4 mRNA thus allowing for its translation to the toxic protein. The new concept here is that this stabilizing PAS is in fact very weak, causing transcription read through and extension to the potent SV40 PAS (in the construction used to generate the previous iDUX4[2.7] mouse) that caused excess DUX4mRNA stabilization and thus translation. So could the Authors please expand a little by providing this background.

Typos:

- Introduction line 7: although major focus of attention DUX4 is not a person, so "DUX4 target genes", not "DUX4's target genes"
- Legend to Fig2D and elsewhere in the text: "6-week-old" not "6 weeks old"
- All figure legends: panels are indicated with capital letters while they aren't on the figures
- The scale bars in histology pictures are either missing or too small.

Reviewer #3 (Remarks to the Author):

Paper review for Nature Communication

Muscle pathology from Stochastic low level DUX4 expression in an FSHD mouse model.
Authors: Bosnakovski D et al.

Since the molecular cause of Fascioscapulohumeral Muscular Dystrophy (FSHD) has been linked to an aberrant expression of the transcription factor DUX4, the investigators are trying to develop an animal model for the disease by loss of expression of DUX4. An altered low level of DUX4 is observed in FSHD patient biopsies. So far the animal model of FSHD is hindered by two limitations; 1) the severity of the disease and 2) the lack of muscle phenotype. The authors developed an FSHD model with conditional titratable expression of DUX4 in skeletal muscle. The muscle degenerative process involves inflammation, with the expansion of fibro-adipogenic progenitors. Also, the model develops a high frequency hearing defect similar to the FSHD patients. The authors believe this new model would be helpful to develop a treatment for FSHD.

The paper is well written and the results support their conclusion. My concern regarding the paper is the fact that numerous groups have attempted to develop FSHD animal models in the past and it is not clear to the reviewer why the current model is better than the other animal models. The author published a paper in 2014 describing another animal model of FSHD and it is unclear how this model is better than the previous one.

Also, FSHD is characterized by having muscles affected at different levels (for example: upper body muscles are more affected than lower body muscles). Therefore, it is not clear whether this new animal model also has skeletal muscles with different disease severity. In Supplemental Figure 2B, the author shows that the muscle mass change between muscle groups but is it really the muscle regeneration, inflammation, and fibrosis level change between muscle? How about the content of FAPs in those different muscle groups? How about looking at angiogenesis and population of progenitor cells that are derived from the blood vessel walls (for example: pericytes, endothelial cells, Adventitia cells, etc.)?

Overall, the paper is well written and well supported by extensive results. My limitation is the fact that a lack of discussion is provided in terms of a comparison between this model and previous animal models of FSHD. One aspect of the disease that is difficult to comprehend is why certain muscle groups are affected while others are not. Finally, a PubMed search on animal models of FSHD led to twenty two papers, so therefore, the novelty of this study is limited at best.

Based on the above comments, I suggest that the paper be revised according to the comments above and resubmitted for publication in Nature Communication.

Responses to specific reviewers' comments:

Reviewer #1 (Remarks to the Author):

Reviewer summary:

Previous attempts at producing a mammalian animal model for FSHD have resulted in animals without a clear skeletal muscle phenotype or animals with a severe, multisystem lethal phenotype. Bosnakovski et al. present data on a transgenic mouse model of a doxycycline-inducible human DUX4 gene (iDUX4pA) integrated into the X chromosome. These transgenic mice survive into adulthood, have a slowly progressive myopathy aggravated by doxycycline induction and have high frequency hearing loss, an extra-muscular feature seen in patients with severe forms of FSHD. The authors claim that this inducible animal model closely mimics

human FSHD, and will be important in exploring the mechanisms of DUX4 induced myopathy and is of relevance in FSHD drug development. The data provided in support of the conclusions are extensive and clear with enough detail to be reproducible.

The iDUX4pA animal model is the first potentially useful FSHD animal model and will be of great interest to the FSHD research community at a time when multiple therapeutic approaches are being actively explored. Although this study does not provide novel insights into disease mechanism, it does confirm experimentally, what earlier investigations have suspected, that DUX4 induction impedes normal muscle regeneration. Below are my specific comments:

Response:

We thank the reviewer for this positive evaluation.

Reviewer comment:

- Although high frequency hearing loss is a feature of FSHD, it occurs in a small percentage of the patients with FSHD who have severe, early onset disease. The claim by the authors that high frequency hearing loss is highly prevalent and highly penetrant is inaccurate.

Response:

We based our wording on the referenced study of Padberg, where hearing tests revealed >50% of individuals had abnormal sonograms. These may not equate to an appreciation of hearing loss on the part of the patient – perhaps that is the reason for the difference of opinion about prevalence. In any case, we have modified our statement to the more neutral “High frequency hearing loss is a non-muscle phenotype associated with FSHD”.

Reviewer comment:

- FSHD muscle histopathology, except for the occasional occurrence of prominent inflammation, is not distinctive. The authors show that with doxycycline induction of a more active and destructive myopathic process, resulting in increased DUX4 mRNA as well as the ability of demonstrate DUX4 in rare myonuclei by IHC. These changes corresponded to a marked increase in Myo1g and Wfdc3, listed as DUX4 targets. However, I could not find references to either Myo1g or Wfdc3 as established DUX4 targets. It would have been very helpful if the authors demonstrated the presence of some of the other DUX4 targets previously established in human FSHD muscle (Yao et al., 2014)

Response:

We initially identified Myo1g and Wfdc3 as DUX4 targets in C2C12, murine myoblast cell line (Bosnakovski et al. EMBO, 2008). They are direct targets of DUX4 and rapidly induced even by low levels of DUX4. Wfdc3 is also described as a DUX4 target and innate immune response gene in the Krom PLoS Genetics 2013 paper. We use them as our standard readouts to evaluate of DUX4 activity. As the reviewer suggested, we have analyzed several more known

DUX4 target genes, including Zscan4c, a homologue of ZSCAN4, a strong target in human cells. This data has been included in Figure 3d.

Reviewer #2 (Remarks to the Author):

Reviewer summary:

Bosnakovski et al present a transgenic mouse (iDUX4pA) with doxycycline-inducible DUX4 expression in skeletal muscles that closely mimics the pathological features of FSHD. This is a major achievement that clearly deserves publication in Nature Communications. Many attempts from several groups had failed up to now at generating an animal model of FSHD that could recapitulate the main characteristics of the human disease. The new iDUX4pA mice present tissue-specific DUX4 expression in skeletal muscles. This expression is tunable with doxycycline and can provide the wide level range observed in FSHD myoblast cultures where it is correlated with different DUX4 gene methylation levels.

There is a major interest in such a model because (i) it will allow for a better understanding of the FSHD pathological mechanism i.e. precisely deciphering the path between the molecular defect and the overt muscle (histo)pathology; (ii) it will provide an invaluable tool for screening or validating putative therapeutic agents (targeting DUX4 expression or pathways downstream of it) that are increasingly being sought for by researchers and pharmaceutical companies. In the previous iDUX4[2.7] mouse model leaky ubiquitous DUX4 expression was highly toxic to male mice. Here, male iDUX4pA mice also show leaky DUX4 expression in the absence of inducer but its very low level strongly reduces the toxicity so they survive for at least 5 weeks.

The mouse characterization has been done thoroughly (weight, muscle function, histology, cell sorting, RNA analyses for DUX4, its target genes, myogenic and fibrogenic genes; muscle regeneration).

Males in the absence of inducer have low leaky DUX4 expression, decreased muscle function with no muscle atrophy but associated with decreased myogenesis markers and increased fibroadipogenic progenitors. This point is of major interest since increased extracellular matrix deposition and fat infiltration are early events in FSHD muscles .

Both males and females have a skin phenotype which is not observed in patients but present high frequency hearing impairment that is typical of patients.

Dose- and time-dependent myopathy (atrophy) develops following DUX4 induction in all mice, with inflammatory infiltrates and fibrosis. This indicates that the selection against cells with the active X chromosome carrying the DUX4 transgene that was observed in the previous iDUX4[2.7] model does not occur here.

Muscle regeneration following cardiotoxin-mediated injury is impaired by DUX4 in a dose-dependent manner.

Response:

We thank the reviewer for this positive summary and evaluation.

Reviewer comment:

Could the Authors comment the following points?

- In non-induced mice where (in which tissues) does the leaky DUX4 expression occur? If indeed it is not possible to immunodetect DUX4 in tissue sections maybe detection of its footprint gene products could be done? (this experiment is not required for the present publication)

Response:

Although not required for the publication, we felt it would be valuable for the paper, so we went ahead and did it, and include data in the revised version of the manuscript (Supplementary Fig.4) . We were able to detect low levels of DUX4 mRNA in almost all tissues from iDUX4pA mice. Since the house keeping genes used for normalization were differently expressed in different tissues it is hard to compare levels of DUX4 expression among analyzed tissues. Most of the analyzed DUX4 target genes did not follow expression of DUX4 in the absence of dox. For example, Myo1g was elevated only in the skin sample and Zscan4c and Mbd3l2 were not detectable at all. It could be that they are cell type-specific targets of DUX4 or in some cases are already highly expressed and low levels of DUX4 do not have a significant effect. Despite the basal detectable DUX4 mRNA expression we were not able to detect DUX4 protein expression by immunostaining or western blot.

Reviewer comment:

- What promoter could be used for this leaky expression? Is it linked to Sp1 binding sites where transcription could occur without a TATAA box?

Response:

The basal expression is coming from the TRE promoter, which is off in the absence of doxycycline or rtTA. Although we used a second generation TRE with reduced leaky expression, as in much of biology, “off” is not equal to zero. We have not investigated the transcript, but we assume that it initiates from the minimal promoter of the TRE, which does in fact have a TATA box.

Reviewer comment:

- In doxycyclin-induced mice the HSA promoter leads to DUX4 expression after differentiation has been initiated thus not in proliferating precursor cells, so a pathogenic function of DUX4 at this early stage could not be detected in this mouse model. What is the nature of the cells with increased DUX4 expression in the regenerating muscles?

Response:

Since the system is active in skeletal actin-positive cells, we believe that the cells with increased DUX4 expression are newly-forming fibers. A comment to this point is included.

Reviewer comment:

- Is there any evidence of DUX4 leaking from skeletal muscle cells to other cell types in this mouse model?

Response:

We have performed a careful analyses of DUX4 expression in muscle sections. We observed DUX4 expressing nuclei under the myofiber lamina, thus we exclude expression in interstitial cells. In addition, majority of the DUX4 expressing cells were MHC positive. These new data are included in Supplementary Fig. 7. However, with the tools that we have it is hard to perfectly characterize all of the DUX4 expressing cells in severely damaged muscle. There is a lot of fragmentation and resorption of myofibers and infiltration of other inflammatory cells as well as proliferation of myoblasts and formation of new small fibers. Without specific markers and good antibodies that we can use for constraining with DUX4 is almost impossible to rule out that DUX4 is not ever expressed in other cell types in this system.

However, because we do not see induction of DUX4 in iDUXpA;HSA-rtTA in other tissues than muscle, we can speculate that DUX4 is induced only in myogenic cells (this new data is in Fig. 3d). This is in agreement with previous characterization of HSA-rtTA mice, where demonstrated the tightness and specificity of the system was demonstrated (Rao et al. 2009).

Reviewer comment:

What is the nature of the non-cell autonomous mechanism suggested in regeneration (discussion)? Could it be linked to the Cxcr4-Sdf1 axis as observed by Dmitriev et al 2016 Oncotargets?

Response:

This would be a topic for subsequent study, but we have now referenced the Dmitriev paper in the discussion section as an example of non cell-autonomous mechanisms in FSHD.

Reviewer comment:

- Previous studies have demonstrated differences in the gene deregulation cascade caused by DUX4 expression when studied in mouse or human cells. This is the case for genes with promoters derived from repeated elements that harbor DUX4 binding sites and are only present in human. In what respect would that explain pathological differences observed in the iDUX4pA

mice and patients?

Response:

We agree that there are differences between the transcriptional profiles of DUX4 in human and mouse cells, and these may result in phenotypic differences. However, our data argues that these differences are not necessary for pathology, and thus the similarities in gene expression profile are where the relevant pathological processes should be investigated. Furthermore, we do not wish to elaborate a complex hypothesis involving repeat elements that might explain differences between pathology in this mouse model and human FSHD as there are other fundamental differences, like the system we are using to confine expression to skeletal muscle, and the regenerative potential of human vs. mouse satellite cells, that are impossible to disentangle. We have added a paragraph of discussion of these points.

Minor comments:

1. Besides the epigenetic conditions the authors should mention in the introduction the genetic background required to develop FSHD i.e. a DUX4 gene with a polyadenylation signal (exon3 PAS) provided to the distal D4Z4 unit by the flanking pLAM sequence. This PAS is unusual in that its sequence is ATTAAA (Dixit et al 2007) but not the standard AATAAA as wrongly stated both in Figure 2a and Supplementary Fig 1. This should be stated in the beginning of the Result section and corrected in the Figures.

Response:

We have added a reference to the allele-specific poly A in the introduction.

We thank the reviewer for pointing out the error in the schematic shown in Fig. 2A. The ATTAAA is commonly used but more frequently subject to alternative usage, thus regulation as described by [E. Beaudoin, S. Freier, J. R. Wyatt, J. M. Claverie, D. Gautheret, Patterns of variant polyadenylation signal usage in human genes. *Genome Res.* 10, 1001 (2000). doi:10.1101/gr.10.7.1001 pmid:10899149]. This may indeed be why it is a weak polyA. We have corrected the figures and included a reference to the alternative usage of the ATTAAA pA.

Minor comment:

2. In the same figures: the first intron is in fact alternatively spliced (Dixit et al 2007; see review Anseau et al *Genes* 2017).

Response:

We have edited the figure to show that the first intron is alternatively spliced by showing both options: including the intron as a solid line and the splice as a dotted line.

Minor comment:

3. The authors have been stumbling for several years on the extreme DUX4 toxicity (even with no doxycycline induction) in their previous iDUX4 [2.7] mouse model, but not the readers, so could they please clarify the background! Lemmers et al 2010 demonstrated the exon3 PAS was required to get stable polyadenylated DUX4 mRNA thus allowing for its translation to the toxic protein. The new concept here is that this stabilizing PAS is in fact very weak, causing transcription read through and extension to the potent SV40 PAS (in the construction used to generate the previous iDUX4[2.7] mouse) that caused excess DUX4mRNA stabilization and thus translation. So could the Authors please expand a little by providing this background.

Response:

We have now provided this background, in the first sentence of the results section.

Reviewer comment:

Typos:

- Introduction line 7: although major focus of attention DUX4 is not a person, so “DUX4 target genes”, not “DUX4’s target genes”
- Legend to Fig2D and elsewhere in the text: “6-week-old” not “6 weeks old”
- All figure legends: panels are indicated with capital letters while they aren’t on the figures
- The scale bars in histology pictures are either missing or too small.

Response:

Thank you for pointing these out. We have corrected them all.

Reviewer #3 (Remarks to the Author):

Paper review for Nature Communication

Muscle pathology from Stochastic low level DUX4 expression in an FSHD mouse model.
Authors: Bosnakovski D et al.

Reviewer summary:

Since the molecular cause of Fascioscapulohumeral Muscular Dystrophy (FSHD) has been linked to an aberrant expression of the transcription factor DUX4, the investigators are trying to develop an animal model for the disease by loss of expression of DUX4. An altered low level of DUX4 is observed in FSHD patient biopsies. So far the animal model of FSHD is hindered by two limitations; 1) the severity of the disease and 2) the lack of muscle phenotype. The authors

developed an FSHD model with conditional titratable expression of DUX4 in skeletal muscle. The muscle degenerative process involves inflammation, with the expansion of fibro-adipogenic progenitors. Also, the model develops a high frequency hearing defect similar to the FSHD patients. The authors believe this new model would be helpful to develop a treatment for FSHD.

The paper is well written and the results support their conclusion. My concern regarding the paper is the fact that numerous groups have attempted to develop FSHD animal models in the past and it is not clear to the reviewer why the current model is better than the other animal models. The author published a paper in 2014 describing another animal model of FSHD and it is unclear how this model is better than the previous one.

Also, FSHD is characterized by having muscles affected at different levels (for example: upper body muscles are more affected than lower body muscles). Therefore, it is not clear whether this new animal model also has skeletal muscles with different disease severity.

Response:

We thank the reviewer for the positive comments regarding the quality of the writing and the results. We discuss the points raised in more detail below, in our response to the last specific reviewer comment, but briefly, we contend that previous models have all had significant problems, and although we do not solve everything, such as why certain muscles are more strongly affected, this new model is a significant step forward in that it generates a slow progressive disease, and is driven by low level, stochastic expression of DUX4.

Reviewer comment:

In Supplemental Figure 2B, the author shows that the muscle mass change between muscle groups but is it really the muscle regeneration, inflammation, and fibrosis level change between muscle? How about the content of FAPs in those different muscle groups? How about looking at angiogenesis and population of progenitor cells that are derived from the blood vessel walls (for example: pericytes, endothelial cells, Adventitia cells, etc.)?

Response:

As the reviewer suggested, we analyzed the content of FAPs, myogenic progenitors, endothelial cells, and macrophages in different muscle groups (TA, gastrocnemius and soleus, quadriceps, pectoralis and triceps). This data is included as Supplementary Figure 3 and described in the Results section. Among all muscle groups, the main difference was consistently in the frequency of the FAPs, as we previously identified on pool sample from the same muscle groups. In addition, small but significant alterations of CD45^{neg} Sca1⁺ CD31⁺ endothelial cells and CD206⁺ macrophages were detected in certain muscle groups. Other cell types were not significantly affected.

Reviewer comment:

Overall, the paper is well written and well supported by extensive results. My limitation is the fact that a lack of discussion is provided in terms of a comparison between this model and previous animal models of FSHD. One aspect of the disease that is difficult to comprehend is why certain muscle groups are affected while others are not. Finally, a PubMed search on animal models of FSHD led to twenty two papers, so therefore, the novelty of this study is limited at best.

Response:

We thank the reviewer for these positive comments regarding the results and the readability of the paper. We point out that this model was not designed with the question of why particular muscle groups are more affected in mind, as we used a generic muscle promoter to express DUX4. The value of this model will be for the study of muscle pathology in response to DUX4, and for the testing of therapeutics targeting the DUX4 message or protein. Regarding novelty, counting the numbers in a PubMed search with the terms animal models + FSHD does not indicate that a generically useful animal model has been generated. If anything, because there are only a handful of papers, it argues that a highly useful animal model is still lacking. We also point out that developing an animal model suitable for testing therapeutics was highlighted as one of the current high priorities for the field in the latest FSHD International Research Consortium Workshop, sponsored by the FSH Society and held in Boston in November 2016, which would not be the case if one existed. Consortium-defined priorities have in the past been influential for FSHD research foundations, the MDA, and the NIH, and indeed, our work has been guided by this consensus priority, therefore we strongly contend that our study represents results of novelty with high interest and relevance to the field.

Reviewer comment:

Based on the above comments, I suggest that the paper be revised according to the comments above and resubmitted for publication in Nature Communication.

Response:

We thank the reviewer for this positive recommendation.

REVIEWERS' COMMENTS:

Reviewer #1 (Remarks to the Author):

This a resubmission of the manuscript after the authors addressed the comments of the three reviewers. Dr. Kyba has responded appropriately to all the comments of all reviewers.

Reviewer #2 (Remarks to the Author):

The Authors have correctly dealt with all the Reviewers' comments both in their answers and in the revised version of the manuscript.

However a few typos should be corrected:

lines

59: add gene "expression"

144: suppress "was"

154: suppress "at"

155: define "CSA"

162: add "in" response

184: add polyA "signal"

186: add pA "signal"

247: clarify the sentence: The SV40 pA signal was deleted from p2Lox-DUX4 20 and "the resulting plasmid introduced by electroporation" into ZX1 inducible 248 cassette exchange mouse ES cells.

One electroporates cells, not DNA

447: add pA "signal"

456-460: add pA "signal"

Fig1 d: correct "females"

Reviewer #3 (Remarks to the Author):

Below are my initial critiques of the paper. All my critiques raised during the first review have been adequately answered by the authors, either through additional experiments or by clarifying some issues in the revised paper. Therefore, I recommend that the paper be accepted, in its current form, for publication in the Journal Nature Communications.

Reviewer #3

Paper review for Nature Communication

Muscle pathology from Stochastic low level DUX4 expression in an FSHD mouse model.

Authors: Bosnakovski D et al.

Since the molecular cause of Fascioscapulohumeral Muscular Dystrophy (FSHD) has been linked to an aberrant expression of the transcription factor DUX4, the investigators are trying to develop an animal model for the disease by loss of expression of DUX4. An altered low level of DUX4 is observed in FSHD patient biopsies. So far the animal model of FSHD is hindered by two limitations; 1) the severity of the disease and 2) the lack of muscle phenotype. The authors developed an FSHD model with conditional titratable expression of DUX4 in skeletal muscle. The muscle degenerative process involves inflammation, with the expansion of fibro-adipogenic

progenitors. Also, the model develops a high frequency hearing defect similar to the FSHD patients. The authors believe this new model would be helpful to develop a treatment for FSHD.

The paper is well written and the results support their conclusion. My concern regarding the paper is the fact that numerous groups have attempted to develop FSHD animal models in the past and it is not clear to the reviewer why the current model is better than the other animal models. The author published a paper in 2014 describing another animal model of FSHD and it is unclear how this model is better than the previous one.

Also, FSHD is characterized by having muscles affected at different levels (for example: upper body muscles are more affected than lower body muscles). Therefore, it is not clear whether this new animal model also has skeletal muscles with different disease severity. In Supplemental Figure 2B, the author shows that the muscle mass change between muscle groups but is it really the muscle regeneration, inflammation, and fibrosis level change between muscle? How about the content of FAPs in those different muscle groups? How about looking at angiogenesis and population of progenitor cells that are derived from the blood vessel walls (for example: pericytes, endothelial cells, Adventitia cells, etc.)?

Overall, the paper is well written and well supported by extensive results. My limitation is the fact that a lack of discussion is provided in terms of a comparison between this model and previous animal models of FSHD. One aspect of the disease that is difficult to comprehend is why certain muscle groups are affected while others are not. Finally, a PubMed search on animal models of FSHD led to twenty two papers, so therefore, the novelty of this study is limited at best.

Based on the above comments, I suggest that the paper be revised according to the comments above and resubmitted for publication in Nature Communication.

REVIEWERS' COMMENTS:

Reviewer #1 (Remarks to the Author):

This a resubmission of the manuscript after the authors addressed the comments of the three reviewers. Dr. Kyba has responded appropriately to all the comments of all reviewers.

RESPONSE: *We thank the reviewer for this positive review.*

Reviewer #2 (Remarks to the Author):

The Authors have correctly dealt with all the Reviewers' comments both in their answers and in the revised version of the manuscript.

However a few typos should be corrected:

lines

59: add gene "expression"

144: suppress "was"

154: suppress "at"

155: define "CSA"

162: add "in" response

184: add polyA "signal"

186: add pA "signal"

247: clarify the sentence: The SV40 pA signal was deleted from p2Lox-DUX4 20 and "the resulting plasmid introduced by electroporation" into ZX1 inducible 248 cassette exchange mouse ES cells. One electroporates cells, not DNA

447: add pA "signal"

456-460: add pA "signal"

Fig1 d: correct "females"

RESPONSE: *We thank the reviewer for pointing out these typos. We have corrected all of them.*

Reviewer #3 (Remarks to the Author):

All my critiques raised during the first review have been adequately answered by the authors, either through additional experiments or by clarifying some issues in the revised paper. Therefore, I recommend that the paper be accepted, in its current form, for publication in the Journal Nature Communications.

RESPONSE: *We thank the reviewer for this positive review.*